# Tumor-Localized Administration of α-GalCer to Recruit Invariant Natural Killer T Cells and Enhance Their Antitumor Activity against Solid Tumors

**DOI:** 10.3390/ijms23147547

**Published:** 2022-07-07

**Authors:** Yan-Ruide Li, Yang Zhou, Matthew Wilson, Adam Kramer, Ryan Hon, Yichen Zhu, Ying Fang, Lili Yang

**Affiliations:** 1Department of Microbiology, Immunology and Molecular Genetics, University of California, Los Angeles, CA 90095, USA; charlie.li@ucla.edu (Y.-R.L.); zzydcat@g.ucla.edu (Y.Z.); mwilson193@g.ucla.edu (M.W.); akramer931@g.ucla.edu (A.K.); ryanhonchenghao@gmail.com (R.H.); yichenzhujohn@g.ucla.edu (Y.Z.); yingfang1913@g.ucla.edu (Y.F.); 2Eli and Edythe Broad Center of Regenerative Medicine and Stem Cell Research, University of California, Los Angeles, CA 90095, USA; 3Jonsson Comprehensive Cancer Center, David Geffen School of Medicine, University of California, Los Angeles, CA 90095, USA; 4Molecular Biology Institute, University of California, Los Angeles, CA 90095, USA

**Keywords:** invariant natural killer T (iNKT) cell, α-GalCer, cancer immunotherapy, solid tumor, melanoma, lung cancer, CD1d, bioluminescence live animal imaging (BLI)

## Abstract

Invariant natural killer T (iNKT) cells have the capacity to mount potent anti-tumor reactivity and have therefore become a focus in the development of cell-based immunotherapy. iNKT cells attack tumor cells using multiple mechanisms with a high efficacy; however, their clinical application has been limited because of their low numbers in cancer patients and difficulties in infiltrating solid tumors. In this study, we aimed to overcome these critical limitations by using α-GalCer, a synthetic glycolipid ligand specifically activating iNKT cells, to recruit iNKT to solid tumors. By adoptively transferring human iNKT cells into tumor-bearing humanized NSG mice and administering a single dose of tumor-localized α-GalCer, we demonstrated the rapid recruitment of human iNKT cells into solid tumors in as little as one day and a significantly enhanced tumor killing ability. Using firefly luciferase-labeled iNKT cells, we monitored the tissue biodistribution and pharmacokinetics/pharmacodynamics (PK/PD) of human iNKT cells in tumor-bearing NSG mice. Collectively, these preclinical studies demonstrate the promise of an αGC-driven iNKT cell-based immunotherapy to target solid tumors with higher efficacy and precision.

## 1. Introduction

Human invariant natural killer T (iNKT) cells are a subset of αβ T lymphocytes that express a semi-invariant CD1d-restricted αβ T cell receptor (TCR) consisting of a conserved TCR α chain (Vα24-Jα18) paired with limited diverse TCR β chains (dominantly Vβ11) [1]. This small subset of T cells is present at a frequency of less than 0.1% in human blood [2]. iNKT cells have recently become a popular focus for cancer therapies due to their ability to directly and indirectly induce tumor cell lysis, playing a key role in the orchestration of both the innate and adaptive immune responses against tumors [1,3,4,5].

Besides expressing iNKT TCR, human iNKT cells also express a variety of natural killer (NK)-activating receptors such as NKG2D, NKp30, NKp44, DNAM-1, and NKR-P1c, which can target NK ligands on tumor or other stressed cells to initiate NK-like killing pathways [2,6]. Upon activation, iNKT cells are able to secrete a large quantity of Th0/Th1 proinflammatory cytokines including interferon-γ (IFN-γ), tumor necrosis factor-α (TNF-α), and interleukin-2 (IL-2), as well as Th2 immunoregulatory cytokines such as IL-4, IL-5, IL-6, IL-10, and IL-13, enabling iNKT cells to participate in the cytotoxic activity and the regulation of an immune response with multiple mechanisms [2,7]. iNKT cells can assist in immune function against tumors by directly killing via Fas–FasL interaction, perforins, and granzyme B; iNKT cells are also able to indirectly assist by activating NK cells, dendritic cells (DCs), macrophages, neutrophils, and lymphocytes [8,9].

Although displaying potent antitumor abilities, iNKT cells are limited in their clinical applications because of their low numbers in human peripheral circulation. Furthermore, these cells have shown evidence of functional exhaustion such as that seen in conventional αβ T cells [9,10]. Efforts to increase cell numbers have led to iNKT cell expansion by alpha-galactosylceramide (α-GalCer or αGC) stimulation. The glycolipid αGC is presented by the nonclassical major histocompatibility complex (MHC) class-I molecule CD1d to the iNKT TCR to effectively stimulate iNKT cells [2,7,11,12]. The CD1d/αGC/iNKT TCR interaction mediates both direct and indirect activation pathways that allow iNKT cells to recognize and attack multiple hematological malignancies and solid tumors [13,14]. In vivo studies have demonstrated that the stimulation of iNKT cells in mice with soluble αGC is able to boost immune response against B16 melanoma; the effects of αGC was further enhanced when presented on dendritic cell vehicles as antigen-presenting cells (APCs) to interact with iNKT cells [15]. These studies have presented various methods of αGC stimulation as potential options to raise iNKT cell numbers to therapeutic levels in order to treat cancers.

Clinical trials focusing on the αGC-mediated activation of iNKT cells have demonstrated effective killing against solid tumors, including non-small cell-lung cancer, head and neck squamous cell carcinomas, and melanoma [14,16,17]. These studies have demonstrated the effective expansion and activation of iNKT cells when stimulated by αGC alone, αGC-pulsed DCs, or other αGC-pulsed APCs [14,16,17]. Our previous preclinical studies utilizing BCMA chimeric antigen receptor (CAR)-engineered and αGC-expanded iNKT cells indicated a feasible safe and tolerable adoptive therapy capable of eliminating multiple myeloma [18,19]. Enhanced tumor killing and more beneficial patient outcomes have been evidenced by αGC-boosted iNKT cell-based therapy; however, in some solid tumors, iNKT cells are still restricted in their antitumor capabilities due to difficulties in infiltrating solid tumors [20].

Here, using humanized NSG mouse models, we show that the tumor-localized administration of a single dose of αGC can effectively recruit iNKT cells into the solid tumor environment in as little as one day while significantly enhancing the antitumor capacity of iNKT cells. These results demonstrate the potential of an αGC-driven iNKT cell-based immunotherapy to more precisely and efficiently target solid tumors.

## 2. Results

### 2.1. Generation and Characterization of Healthy Donor Peripheral Blood Mononuclear (PBMC)-Derived iNKT (PBMC-iNKT) Cells

Human PBMCs were collected from healthy donors and used to generate PBMC-iNKT cells. Using magnetic-activated cell sorting (MACS), iNKT cells were isolated and enriched from PBMCs and then stimulated with α-galactosylceramide (αGC)-loaded, donor-matched irradiated PBMCs (Figure 1A). Since T cells require interleukin 7 (IL-7) and NK cells require interleukin 15 (IL-15) for their expansion and activation [21], these two cytokines were supplemented in the PBMC-iNKT cell culture (Figure 1A). In the presence of αGC, PBMC-iNKT cell proliferation is vigorous compared to that in the absence of αGC (Figure 1B). Flow cytometry demonstrated that with the stimulation of αGC, the PBMC-iNKT cell population exceedingly increased from less than 1% to over 99% (Figure 1C); here, the iNKT T-cell receptor (TCR) and αβ conventional TCR were used to identify iNKT cells. Different from PBMC-derived αβ conventional T cells (PBMC-Tc), which have two cell subpopulations (CD4 single-positive (SP) helper T cells and CD8 SP cytotoxic T cells), PBMC-iNKT cells contain CD4 SP, CD8 SP, and double-negative (DN) cell populations (Figure 1C). Furthermore, compared to PBMC-Tc cells, PBMC-iNKT cells express higher levels of NK-activating receptors such as CD161, NKG2D and DNAM-1, T cell memory marker CD45RO, and T and NK cell effector marker CD69 (Figure 1C).

### 2.2. Tumor Targeting of PBMC-iNKT Cells

To evaluate the antitumor capacity of PBMC-iNKT cells, we used an established in vitro human melanoma A375-CD1d-FG and lung cancer H292-CD1d-FG tumor cell killing assays with or without the addition of αGC; PBMC-Tc cells were included as controls (Figure 2A). The human melanoma A375-CD1d-FG cell line and the human lung cancer H292-CD1d-FG cell line were generated by engineering their parental cell lines to overexpress human CD1d and the firefly luciferase (Fluc) and enhanced green fluorescence protein (EGFP) dual-reporters (FG) to allow for the convenient monitoring of the tumor cells through luciferase assays or flow cytometry (Figure 2B). The flow cytometry analysis of both cell lines confirmed the overexpression of human CD1d (Figure 2C). PBMC-iNKT or PBMC-T cells were mixed with tumor cells and incubated for 24 h. After 24 h, D-luciferin was added to react with Fluc from the FG dual-reporter, resulting in fluorescent signals of varying intensities representative of the amount of live tumor cells in the coculture (Figure 2A). Compared to conventional PBMC-Tc cells, PBMC-iNKT cells exhibited a significant antitumor capacity across both cell lines (Figure 2D). Furthermore, the addition of αGC further enhanced the tumor cell killing efficacy of PBMC-iNKT cells but not of PBMC-T cells, indicating an antitumor function mediated by iNKT TCR (Figure 2D). The flow cytometry analysis of PBMC-iNKT cells revealed enhanced cytotoxic function; compared to just PBMC-iNKT cells, PBMC-iNKT cells cocultured with tumor cells displayed greater levels of CD69, perforin, and granzyme B, while PBMC-iNKT cells cocultured with tumor cells and treated with αGC displayed greater levels of CD69, perforin, and granzyme B compared to both PBMC-iNKT cells cocultured with or without tumor cells (Figure 2E,F).

Next, we evaluated the in vivo antitumor capacity of PBMC-iNKT cells using a human lung cancer xenograft NSG (NOD.Cg-Prkdc^scid^Il2rg^tm1Wjl^/SzJ) mouse model. H292-CD1d-FG tumor cells were subcutaneously inoculated into NSG mice on day 0. Time-course live animal bioluminescence imaging (BLI) for monitoring tumor size began on day 2, followed by the intravenous injection of a PBS control (denoted as Vehicle) or PBMC-iNKT cells with or without the tumor-localized injection of αGC on day 3 (Figure 3A). The tumor-localized administration of a single dose of αGC significantly enhanced the antitumor ability of PBMC-iNKT cells, as evidenced by BLI monitoring (Figure 3B,C), tumor size measurements (Figure 3D), and the terminal analysis of tumor weights (Figure 3E). These studies were also performed using an A375-CD1d-FG human melanoma xenograft NSG mouse model, with comparable results (Figure 4A–E).

### 2.3. Pharmacokinetics/Pharmacodynamics (PK/PD) Study of PBMC-iNKT Cells

In order to study the biodistribution and PK/PD of PBMC-iNKT cells in the human tumor xenograft mouse model, we generated FG-labeled PBMC-iNKT (PBMC-iNKT-FG) cells. These cells were cultured from non-activated PBMC-iNKT cells isolated from human PBMCs using MACS; stimulated with αGC, IL-7, and IL-15; and then transduced with a Lenti/FG lentiviral vector to overexpress the FG dual-reporter for convenient BLI monitoring (Figure 5A). Here, we used an in vivo A375-CD1d human melanoma xenograft NSG mouse model, wherein A375-CD1d tumor cells were subcutaneously inoculated into the left side of the experimental NSG mice on day 0, followed by the tumor-localized injection of αGC and the intravenous injection of PBMC-iNKT-FG cells on day 3 (Figure 5B). A subset of NSG mice intravenously injected with only PBMC-iNKT cells was used as a control (Figure 5B).

BLI images from the ventral side of experimental mice showed similar PK/PD of iNKT cells in mice injected with PBMC-iNKT-FG cells alone to that of mice injected with PBMC-iNKT-FG cells, tumor cells, and αGC (Figure 5D,F and Appendix A): initially, the majority of PBMC-iNKT-FG cells accumulated in the lung and with minimal distribution in the intestines on the injection day (day 0); these iNKT cells migrated to the liver and bone marrow (BM) in the following 5 days (Figure 5C,D). The BLI intensity of PBMC-iNKT-FG cells gradually decreased during the first week; after one week, PBMC-iNKT-FG cells expanded and were distributed in multiple tissues including the lung, liver, spleen, BM, intestine, and lymph nodes (Figure 5C,D). The BLI signal from the tail indicated that PBMC-iNKT-FG cells actively circulated in the peripheral blood of the experimental mice (Figure 5D).

However, BLI images from the left side of experimental mice revealed higher PBMC-iNKT cell loads in mice injected with PBMC-iNKT-FG cells, tumor cells, and αGC compared to mice injected with only PBMC-iNKT-FG cells (Figure 5E–H and Appendix A). In the mice injected with PBMC-iNKT-FG cells only, these cells remained distributed in the lung, liver, spleen, BM, intestine, and periphery blood (Figure 5C,E). In the mice injected with A375-CD1d tumor cells and PBMC-iNKT-FG cells, the tumor-localized injection of αGC efficiently recruited PBMC-iNKT-FG cells to tumor sites; the cell signal from the tumor sites increased through the first two weeks, reached a peak around day 13, and then slowly decreased. Meanwhile, PBMC-iNKT-FG cell loads in tissues other than the tumor were relatively similar to those in experimental mice injected with PBMC-iNKT-FG cells only (Figure 5H). In the course of PBMC-iNKT-FG cell treatment, we also evaluated cytokine production in mouse sera, including human IFN-γ and TNF-α (Appendix A). PBMC-iNKT-FG cells could produce high levels of IFN-γ and TNF-α, and the inoculation of A375-CD1d tumor cells and the tumor-localized injection of αGC could further enhance cytokine production by PBMC-iNKT cells (Appendix A). The terminal analysis of the various tissues revealed the distribution of PBMC-iNKT-FG cells: one month after PBMC-iNKT-FG cell injection, these cells were distributed in the spleen, pancreas, lung, liver, bone marrow, reproductive organs, fat tissues, kidneys, and intestine; in the mice injected with PBMC-iNKT-FG cells, tumor cells, and αGC, we observed the prolonged accumulation of PBMC-iNKT cells in the tumor (Figure 5I). These results indicated that the tumor-localized administration of a single dose of αGC effectively recruited PBMC-iNKT cells into the solid tumor environment in as little as one day while significantly enhancing the persistence and antitumor capacity of PBMC-iNKT cells.

## 3. Discussion

The development of iNKT cell-based therapies has become an area of significant interest in recent years; multiple relevant clinical trials have been conducted to determine both the efficacy and safety profile of iNKT cell engraftment in human patients [14]. These trials have reported promising antitumor responses in patients afflicted with a wide range of malignancies including myeloma, melanoma, colon carcinoma, non-small cell lung cancer, and head and neck cancer [22,23,24,25,26,27]. The adoption of iNKT cell-based therapy was found to be markedly safe and well-tolerated in patients, with no contraindicating symptomatology observed [14]. Therapeutic efficacy was correlated with the number and function of infiltrating iNKT cells, APC presentation, and tumor CD1d expression. Preliminary therapies tailored to stimulate endogenous iNKT cells employed αGC, an iNKT-specific glycosphingolipid ligand with previously observed anti-tumor effects in murine models [28]. While single-pulse intravenous doses of αGC could induce a transient spike in proinflammatory cytokines (e.g., TNF-α, IFNγ, and IL-12), these were insufficient to induce iNKT proliferation, indicating the need for alternative administration routes. Subsequent clinical trials investigated antigen-presenting cells (i.e., dendritic cells (DCs)) pulsed with αGC and intravenously injected for the activation of peripheral iNKT cells [29,30,31]. DC-mediated αGC presentation transiently expanded peripheral iNKT cell levels, leading to rapid iNKT tumor infiltration, proinflammatory cytokine profile in the serum, and the adjuvant stimulation of natural killer (NK) cells and cytotoxic lymphocytes (CTLs) [30,31]. These results suggest the powerful anti-tumor potential of iNKT cell activation; however, variability in endogenous iNKT cell quality, especially in immune-exhausted cancer patients, has elicited a transition towards therapies engrafting PBMC-iNKT cells activated using in vitro αGC stimulation [32]. The adoptive transfer of PBMC-iNKT cells displayed similar immunostimulatory effects as the in vivo modulation of mature iNKT cells but with a greater efficacy for late stage cancer patients [33,34,35]. Based on the success of these clinical trials, in this study, we investigated a combinatory therapy of PBMC-iNKT engraftment and tumor-localized αGC inoculation to promote high-efficacy tumor-targeting for the treatment of solid cancers.

One of the barriers our proposed platform addresses is the difficulty of recruiting peripheral iNKT cells to the tumor site [36]. Tumor signaling induces the downregulation of iNKT cell IFN-γ secretion relative to immunosuppressive IL-4 cytokine, suppressing tumor infiltration to evade iNKT anti-tumor cytotoxicity [37]. As an iNKT-specific activating ligand, secondary αGC administration can remediate iNKT inactivation and restore tumor killing efficacy [28,38]. αGC has not only been established to restore deficient iNKT cytotoxicity but also has potential as an iNKT-recruiting chemokine from the specific affinity between these two entities. In this study, we explored the potential of αGC as a chemoattractant through single-dose, tumor-localized αGC injection at the site of subcutaneous H292 and A375 cell line tumors (Figure 3 and Figure 4). Compared to the mice administered a PBS injection control or without αGC stimulation, the PBMC-iNKT αGC group displayed the sustained recruitment of iNKT cells to the tumor site and a significant reduction in tumor burden (Figure 3B–E and Figure 4B–E). The ability to attract iNKT cells to the tumor site provides a powerful tool to maximize the efficacy of iNKT cell-based therapies and precisely restrict iNKT cell biodistribution to the tumor site. While repeated pulses of tumor-localized αGC could potentially maximize iNKT cell recruitment, thereby enhancing antitumor capacity, previous studies suggest that overexposure may instead anergize the iNKT recognition of α-GalCer. Anergic-state iNKT cells display reduced persistence and suppressed Th2 immune response, thus dampening their antitumor effect in patients [39]. Therefore, the proper titration of αGC dosage and window is necessary for the efficient and safe adoption of iNKT cell-based cancer immunotherapy.

Through the adoption of chimeric antigen receptor (CAR)-engineered iNKT (CAR-iNKT) therapy, the anti-tumor capacity of iNKT cells is further enhanced through the precise targeting of tumor-specific molecules [40,41]. The reliance of PBMC-iNKT cells‘ cytotoxic recognition on antigen-presenting protein CD1d heavily restricts their efficacy; CD1d is limited to presenting glycolipid antigens, and multiple cancers downregulate CD1d expression to evade iNKT killing [42]. On the other hand, the engineered CAR construct consists of a synthetic, extracellular single-chain variable fragment (scFv) derived from a recombinant antibody to recognize antigens without the need for APC presentation [43]. A transmembrane hinge region links the scFv region to a customized intracellular domain composed of T-cell-signaling moieties (e.g., CD3ζ and CD28) depending on the generation of the CAR. Preliminary CAR-iNKT therapies have employed various CAR-targeting antigens associated with cancer, including CD19 [44], B cell maturation antigen (BCMA) [45], and GD2 [46] for the treatment of B cell lymphoma, multiple myeloma, and neuroblastoma tumors, respectively. In comparison to their CAR-T counterparts, CAR-iNKT therapy has displayed more robust proliferation, the elevated secretion of proinflammatory cytokines, and enhanced tumor-targeting and cytotoxicity [44,45,46]. In all three cases, CAR targeting enhanced iNKT anti-tumor effects to a greater extent than the respective CAR-T cell therapies, demonstrating the potential of CAR-iNKT as a powerful platform for cell-based immunotherapy. By combining tumor-localized αGC administration and CAR engineering on PBMC-iNKT cells, we could potentially achieve rapid iNKT recruitment to solid tumors and an enhanced iNKT antitumor capacity, providing a new strategy for current iNKT cell-based cancer therapy.

Another barrier restricting the adoption of PBMC-iNKT therapy is the low quantity of endogenous iNKT cells for collection in peripheral circulation, approximately only 0.001–1% of immune cells in the blood [12,47]. To overcome this, we investigated the generation of hematopoietic stem cell (HSC)-derived iNKT (HSC-iNKT) cells to produce a stable iNKT cell repertoire for therapeutic use. The transduction of the human iNKT TCR αβ gene could potential induce HSC-iNKT differentiation, and an initial proof-of-concept of this method was accomplished using an in vivo humanized BLT mouse model [16]. The cell product harvested from transduced BLT mice displayed a significant increase in the HSC-iNKT population, and their anti-tumor capacity was confirmed against an MM.1S human multiple myeloma cell line using in vitro and in vivo models. However, to produce a more massively scalable platform for HSC-iNKT generation, a modified ex vivo artificial thymic organoid (ATO) culture system was designed to grow HSCs [18,19]. While the ATO platform was designed for the in vitro generation of conventional T cells from HSCs using a 3D stromal cell matrix, iNKT TCR transduction combined with αGC stimulation successfully produced a high-yield and high-purity HSC-iNKT cell product [18,19]. These cells retained proper iNKT morphology and functional profile, showing significant anti-tumor capacity against multiple human solid tumor cell lines [18,19]. The capacity of HSC-iNKT to tolerate CAR engineering was also evaluated against a CD1d-engineered MM.1S cell line, wherein BCMA-targeting CAR (BCAR)-engineered HSC-iNKT (BCAR-iNKT) cells showed enhanced cytotoxicity in comparison to mock-transduced HSC-iNKT cells [18,19]. The ATO HSC-iNKT platform holds great potential to generate unlimited amounts of CAR-iNKT cells armed with powerful antitumor efficacy that can also be recruited to tumor sites with localized αGC administration for the highly effective treatment of cancer.

## 4. Materials and Methods

### 4.1. Lentiviral Vectors and Transduction

The lentivector and lentivirus were generated from a parental lentivector pMNDW, as previously described [16]. The Lenti/CD1d vector was constructed by inserting a synthetic gene encoding human CD1d into pMNDW; the Lenti/FG vector was constructed by inserting a synthetic bicistronic gene encoding Fluc-F2A-EGFP into pMNDW. The synthetic gene fragments were obtained from GenScript (Piscataway, NJ, USA) or IDT (Coralville, IA, USA). 293T cells were used to generate lentiviruses following a standard calcium precipitation protocol and an ultracentrifugation concentration protocol, as previously described [1].

### 4.2. Cell Culture Media and Reagents

α-Galactosylceramide (αGC, KRN7000) was purchased from Avanti Polar Lipids (Alabaster, AL, USA). Recombinant human IL-2, IL-7, and IL-15 were purchased from PeproTech (Hamburg, Germany).

RPMI1640 and DMEM cell culture media were purchased from Corning Cellgro (Manassas, VA, USA). Fetal bovine serum (FBS) and beta-mercaptoethanol (β-ME) were purchased from Sigma (St. Louis, MO, USA). Medium supplements, including penicillin–streptomycine–glutamine (P/S/G), MEM non-essential amino acids (NEAA), HEPES Buffer Solution, and sodium pyruvate, were purchased from GIBCO (Waltham, MA, USA). Normocin was purchased from InvivoGen (San Diego, CA, USA). The complete lymphocyte culture medium (denoted as the C10 medium) comprised RPMI 1640 supplemented with FBS (10% *v*/*v*), P/S/G (1% *v*/*v*), MEM NEAA (1% *v*/*v*), HEPES (10 mM), sodium pyruvate (1 mM), β-ME (50 μM), and Normocin (100 μg/mL). The adherent cell culture medium (denoted as the D10 medium) comprised DMEM supplemented with FBS (10% *v*/*v*), P/S/G (1% *v*/*v*), and Normocin (100 μg/mL).

### 4.3. Tumor Cell Lines

Human melanoma cell line A375 and human lung cancer cell line H292 were purchased from the American Type Culture Collection (ATCC). A375 and H292 cells were cultured in the D10 medium. To generate stable tumor cell lines overexpressing human CD1d and/or firefly luciferase and enhanced green fluorescence protein (Fluc–EGFP) dual-reporters, lentiviral vectors encoding the intended gene(s) were used to transduce the parental tumor cell line [18]. At 72 h after lentivector transduction, engineered tumor cells were sorted using flow cytometry. Three stable tumor cell lines were generated for this study: A375-CD1d, A375-CD1d-FG, and H292-CD1d-FG.

### 4.4. Mice

NOD.Cg-Prkdc^SCID^Il2rg^tm1Wjl^/SzJ (NOD/SCID/IL-2Rγ^−/−^, NSG) mice were maintained in the animal facilities at the University of California, Los Angeles (UCLA). Six-to-ten-week-old mice were used for all experiments unless otherwise indicated. All animal experiments were approved by the Institutional Animal Care and Use Committee of UCLA.

### 4.5. Antibodies and Flow Cytometry

All flow cytometry stains were performed in PBS for 15 min at 4 °C. Antibody staining was performed at a dilution according to the manufacturer’s instructions. Fluorochrome-conjugated antibodies specific for human TCRαβ (Clone I26, PB or PE-Cy7-conjugated, 1:50), CD4 (Clone OKT4, FITC or PE-Cy7-conjugated, 1:500), CD8 (Clone SK1, PerCP or APC-Cy7-conjugated, 1:500), CD45RO (Clone UCHL1, PerCP-conjugated, 1:5000), CD161 (Clone HP-3G10, APC or APC-Cy7-conjugated, 1:100), CD1d (Clone 51.1, APC or FITC-conjugated, 1:50), NKG2D (Clone 1D11, APC or PE-Cy7-conjugated, 1:50), DNAM-1 (Clone 11A8, APC or PE-Cy7-conjugated, 1:50), CD69 (Clone FN50, PerCP-conjugated, 1:50), granzyme B (Clone QA16A02, APC or FITC-conjugated, 1:5000), and perforin (Clone dG9, PE-Cy7-conjugated, 1:50) were purchased from BioLegend (San Diego, CA, USA); fluorochrome-conjugated antibodies specific for mouse IgG2b, κ Isotype Ctrl (Clone MPC-11), were purchased from BioLegend; fluorochrome-conjugated antibodies specific for TCR Vα24-Jβ18 (Clone 6B11, PE-conjugated, 1:5) were purchased from BD Biosciences (San Jose, CA, USA). Fixable Viability Dye e506 was purchased from Affymetrix eBioscience (San Diego, CA, USA). Intracellular cytokines were stained using a Cell Fixation/Permeabilization Kit (BD Biosciences). A MACSQuant Analyzer 10 flow cytometer (Miltenyi Biotech, Auburn, CA, USA) was used to perform flow cytometry, with data analyzed by FlowJo software version 9.

### 4.6. Enzyme-Linked Immunosorbent Assay (ELISA)

Mouse sera were collected for cytokine ELISA analysis following a standard protocol from BD Biosciences. The coating and biotinylated antibodies for the detection of human IFN-γ and TNF-α (coating antibody, catalog no. 551220 and no. 551221, respectively; biotinylated detection antibody, catalog no. 554511 and no. 554550, respectively) were purchased from BD Biosciences. The streptavidin–horseradish peroxidase (HRP) conjugate (catalog no. 18410051) was purchased from Invitrogen. Human IFN-γ and TNF-α standards (catalog no. 29-8319-65 and no. 29-8329-65, respectively) were purchased from eBioscience. The 3,3′,5,5′-tetramethylbenzidine (TMB; catalog no. 51200048) substrate was purchased from Kirkegaard & Perry Laboratories (KPL, Gaithersburg, MD, USA). The absorbance at 450 nm was measured using an Infinite M1000 microplate reader (Tecan, Morrisville, NC, USA).

### 4.7. In Vitro Generation of Peripheral Blood Mononuclear (PBMC)-Derived Conventional αß T (PBMC-Tc) Cells

Healthy donor-derived PBMCs were provided by the UCLA/CFAR Virology Core Laboratory without identification information under federal and state regulations. These PBMCs were stimulated with CD3/CD28 T-activator beads (Thermo Fisher Scientific, Waltham, MA, USA) and cultured in the C10 medium supplemented with human IL-2 (20 ng/mL) for 2–3 weeks to generate PBMC-Tc cells, following the manufacturer’s instructions.

### 4.8. In Vitro Generation and Activation of PBMC-Derived iNKT (PBMC-iNKT) Cells

PBMCs were MACS-sorted via anti-iNKT microbead (Miltenyi Biotech) labeling to enrich iNKT cells [48]. Cells were then stimulated with donor-matched irradiated αGC-PBMCs at a ratio of 1:1 and cultured in the C10 medium supplemented with human IL-7 (10 ng/mL) and IL-15 (10 ng/mL) for 2–3 weeks. αGC-PBMCs were generated by culturing PBMCs in the C10 medium with the addition of αGC (5 μg/mL) for 1 h and irradiating PBMCs at 6000 rads. The resulting PBMC-iNKT cells were purified using FACS or MACS via human iNKT TCR antibody (6B11) staining. To generate stable PBMC-iNKT-FG cells, lentiviral vectors encoding FG genes were used to transduce PBMC-iNKT cells. At 72 h after lentivector transduction, engineered PBMC-iNKT cells were stained using flow cytometry to confirm the FG expression. The resulting PBMC-iNKT-FG cells were purified using FACS or MACS staining.

### 4.9. Cell Phenotype Study

The phenotypes of activated and non-activated PBMC-iNKT and PBMC-Tc cells were analyzed using flow cytometry. The phenotypes of these cells were studied by analyzing cell surface markers including co-receptors (CD4 and CD8), NK cell marker (CD161), memory/effector T cell marker (CD45RO and CD69), and NK-activating receptors (NKG2D and DNAM-1).

### 4.10. In Vitro Tumor Cell Killing Assay

A375-CD1d-FG or H292-CD1d-FG tumor cells (1 × 10^4^ cells per well) were cocultured with PBMC-iNKT cells or PBMC-Tc cells (ratios of 1:0, 1:1, 1:2, 1:5, and 1:10) in Corning 96-well Clear Bottom Black plates for 24 h in the C10 medium with or without the addition of αGC (100 ng/mL). At the end of the culture, live tumor cells were quantified by adding D-luciferin (150 μg/mL; Caliper Life Science, Hopkinton, MA, USA) to cell cultures and reading out luciferase activities utilizing an Infinite M1000 microplate reader (Tecan, Morrisville, NC, USA) according to the manufacturer’s instructions. The CD69, Perforin, and Granzyme B expression levels of PBMC-iNKT cells were analyzed using flow cytometry.

### 4.11. Bioluminescence Live Animal Imaging (BLI)

BLI was performed using an IVIS 100 imaging system (Xenogen/PerkinElmer). Live animal imaging was acquired 5 min after the intraperitoneal injection of D-luciferin. For tumor load measurement, 100 μL of D-luciferin was injected. For iNKT cell load measurement, 300 μL of D-luciferin was injected. In some experiments, the iNKT cell load in multiple tissues was measured. At 5 min after the 1 mL injection of D-luciferin, experimental mice were euthanized and the tissues (i.e., tumor, spleen, pancreas, lung, liver, bone marrow, reproductive organs, fat tissues, kidney, heart, muscle, brain, and intestine) were collected. Imaging results were analyzed using Living Imaging 2.50 software (Xenogen, Alameda, CA, USA).

### 4.12. In Vivo Antitumor Efficacy Study: H292 Human Lung Cancer Xenograft NSG Mouse Model

NSG mice were subcutaneously inoculated with 1 × 10^6^ H292-CD1d-FG cells (day 0) and were allowed to develop lung cancer over the course of about 5 weeks. Three days post-tumor inoculation (day 3), mice received the i.v. injection of vehicle (PBS) or 1 × 10^7^ PBMC-iNKT cells with or without the administration of tumor-localized αGC (10 μg/mL; 100 μL/mouse). Over time, tumor loads in experimental animals were monitored starting from day 2 by measuring total body luminescence using BLI (shown as TBL p/s) and by measuring tumor volume using a Fisherbrand^TM^ Traceable^TM^ digital caliper (Thermo Fisher Scientific). Tumor volume was calculated as width^2^ × length × 0.52 (mm^3^). At around week 5, mice were terminated. Solid tumors were retrieved, weighted using a PA84 precision balance (Ohaus), and then compared.

### 4.13. In Vivo Antitumor Efficacy Study: A375 Human Melanoma Xenograft NSG Mouse Model

NSG mice were subcutaneously inoculated with 1 × 10^6^ A375-CD1d-FG cells (day 0) and were allowed to develop melanoma over the course of about 3 weeks. Three days post-tumor inoculation (day 3), mice received the i.v. injection of vehicle (PBS) or 1 × 10^7^ PBMC-iNKT cells with or without the administration of tumor-localized αGC (10 μg/mL; 100 μL/mouse). Over time, tumor loads in experimental animals were monitored starting from day 2 by measuring total body luminescence using BLI (shown as TBL p/s) and by measuring tumor volume using a Fisherbrand^TM^ Traceable^TM^ digital caliper (Thermo Fisher Scientific). Tumor volume was calculated as width^2^ × length × 0.52 (mm^3^). At approximately week 3, mice were terminated. Solid tumors were retrieved, weighted using a PA84 precision balance (Ohaus), and then compared.

### 4.14. In Vivo Pharmacokinetics/Pharmacodynamics (PK/PD) Study of PBMC-iNKT Cells: A375 Human Melanoma Xenograft NSG Mouse Model

NSG mice were subcutaneously inoculated with 1 × 10^6^ A375-CD1d cells (day 0). Three days post-tumor inoculation (day 3), these mice received the i.v. injection of 1 × 10^7^ PBMC-iNKT-FG cells with the administration of tumor-localized αGC (10 μg/mL; 100 μL/mouse). NSG mice injected with PBMC-iNKT-FG cells alone were included as controls. Over time, iNKT cell loads in experimental animals were monitored from both the ventral and left side views using BLI. iNKT cell loads in tumors were specifically measured, and iNKT cell loads in other tissues except tumors were calculated by subtracting the iNKT cell load in tumors from that in the total body of experimental mice. At day 35, mice were terminated, multiple tissues were collected, and iNKT cell loads in tissues were measured using BLI.

### 4.15. Statistical Analysis

Prism 6 software (GraphPad, San Diego, CA, USA) was utilized for all statistical analyses. Pairwise comparisons were performed with a 2-tailed Student’s *t* test. Multiple comparisons were performed with an ordinary 1-way ANOVA followed by Tukey’s multiple comparisons test. Unless otherwise indicated, data are presented as the mean ± SEM. In all figures and figure legends, “n” represents the number of samples or animals used in the indicated experiments. A *p*-value of less than 0.05 was considered significant. ns means not significant; * *p* < 0.05; ** *p* < 0.01; *** *p* < 0.001; **** *p* < 0.0001.

## 5. Conclusions

This study provides a new strategy for iNKT cell-based cancer immunotherapy. In multiple solid tumor models, by administrating a single dose of tumor-localized α-GalCer, human iNKT cells were rapidly recruited into solid tumors in as little as one day and demonstrated an improved tumor suppression capacity. This approach is a promising improvement for iNKT cell therapy to treat solid tumors and lays a foundation for the translational and clinical development.

## Figures and Tables

**Figure 1 ijms-23-07547-f001:**
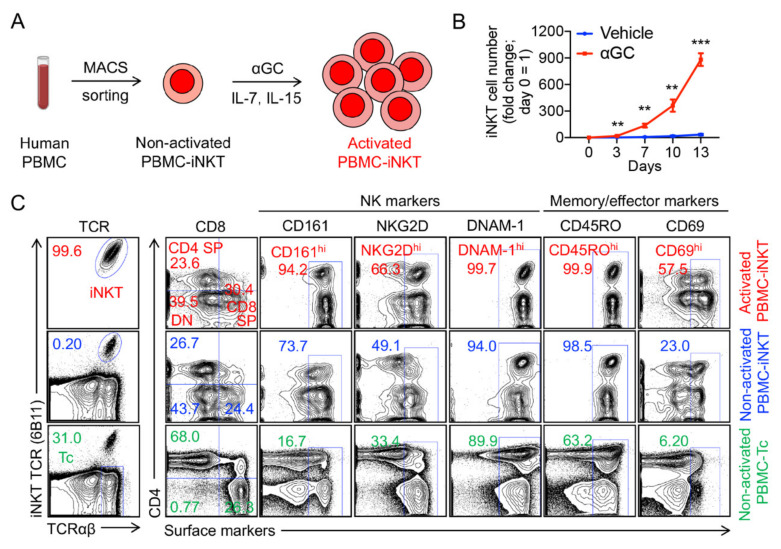
Generation of healthy donor peripheral blood mononuclear (PBMC)-derived iNKT (PBMC-iNKT) cells. (**A**) Experimental design to generate PBMC-iNKT cells in vitro. MACS, magnetic-activated cell sorting; αGC, α-galactosylceramide. (**B**) PBMC-iNKT cell growth curve post-antigen stimulation. PBMC-iNKT cells were cultured for 13 days in the presence or absence of αGC (denoted as αGC or Vehicle, respectively; *n* = 3). (**C**) FACS detection of surface markers on PBMC-iNKT cells. Non-activated PBMC-iNKT cells and non-activated PBMC-Tc cells were included as controls. Representative of three experiments. Data are presented as the mean ± SEM. ** *p* < 0.01; *** *p* < 0.001 by Student’s *t* test.

**Figure 2 ijms-23-07547-f002:**
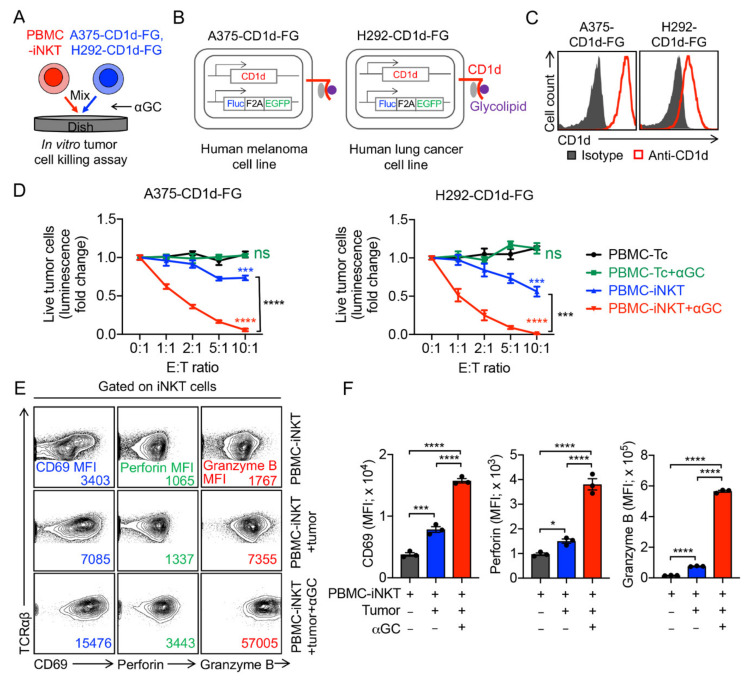
In vitro tumor cell killing efficacy of PBMC-iNKT cells. (**A**) Experimental design. Human melanoma A375-CD1d-FG and lung cancer H292-CD1d FG were studied. The two cell lines were generated by engineering parental cell lines to overexpress human CD1d, as well as firefly luciferase and enhanced green fluorescence protein (FG) dual-reporters. (**B**) Schematics showing the engineered A375-CD1d-FG and H292-CD1d-FG cell lines. (**C**) FACS detection of CD1d on A375-CD1d-FG and H292-CD1d-FG cells. (**D**) Tumor killing data at 24 h (*n* = 4). (**E**) FACS characterization of CD69, perforin, and granzyme B expression of PBMC-iNKT cells. (**F**) Quantification of (**E**) (*n* = 3). Representative of 3 experiments. Data are presented as the mean ± SEM. ns, not significant, * *p* < 0.05; *** *p* < 0.001; **** *p* < 0.0001 by one-way ANOVA.

**Figure 3 ijms-23-07547-f003:**
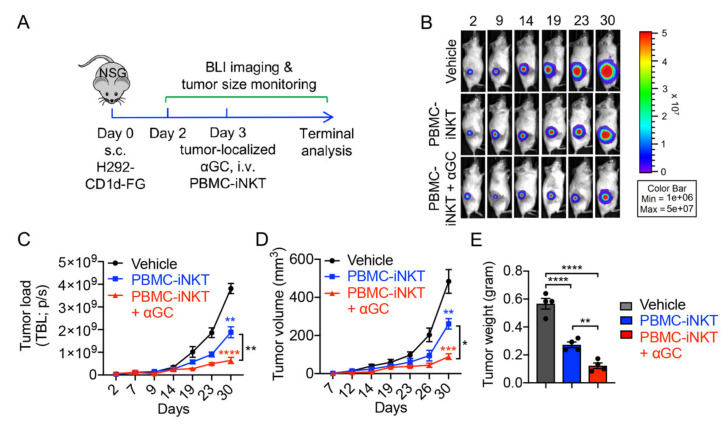
In vivo anti-tumor efficacy of PBMC-iNKT cells in an H292-CD1d-FG human lung cancer xenograft NSG mouse model. (**A**) Experimental design. BLI, live animal bioluminescence imaging; s.c., subcutaneous injection; i.v., intravenous injection. (**B**) BLI images showing tumor loads in experimental mice over time. (**C**) Quantification of (**B**) (*n* = 4). (**D**) Tumor size measurements over time (*n* = 4). (**E**) Tumor weight measurement on day 33 (*n* = 4). Representative of 2 experiments. Data are presented as the mean ± SEM. ns, not significant, * *p* < 0.05; ** *p* < 0.01; *** *p* < 0.001; **** *p* < 0.0001 by one-way ANOVA.

**Figure 4 ijms-23-07547-f004:**
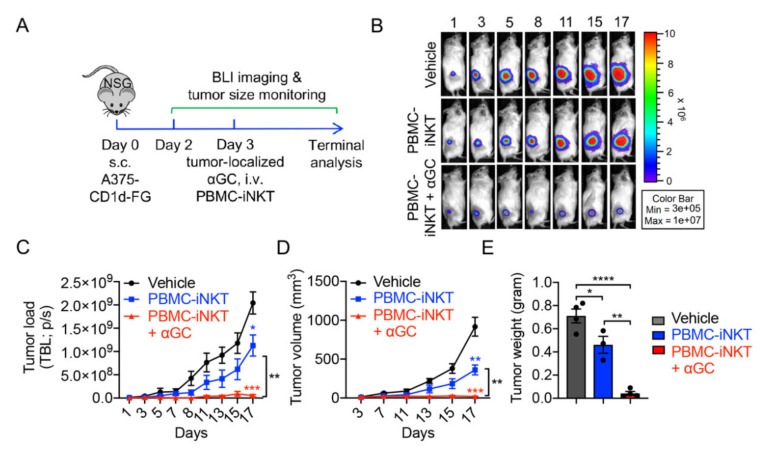
In vivo anti-tumor efficacy of PBMC-iNKT cells in an A375-CD1d-FG human melanoma xenograft NSG mouse model. (**A**) Experimental design. (**B**) BLI images showing tumor loads in experimental mice over time. (**C**) Quantification of (**B**) (*n* = 3–4). (**D**) Tumor size measurements over time (*n* = 3–4). (**E**) Tumor weight measurement on day 20 (*n* = 3–4). Representative of 2 experiments. Data are presented as the mean ± SEM. ns, not significant, * *p* < 0.05; ** *p* < 0.01; *** *p* < 0.001; **** *p* < 0.0001 by one-way ANOVA.

**Figure 5 ijms-23-07547-f005:**
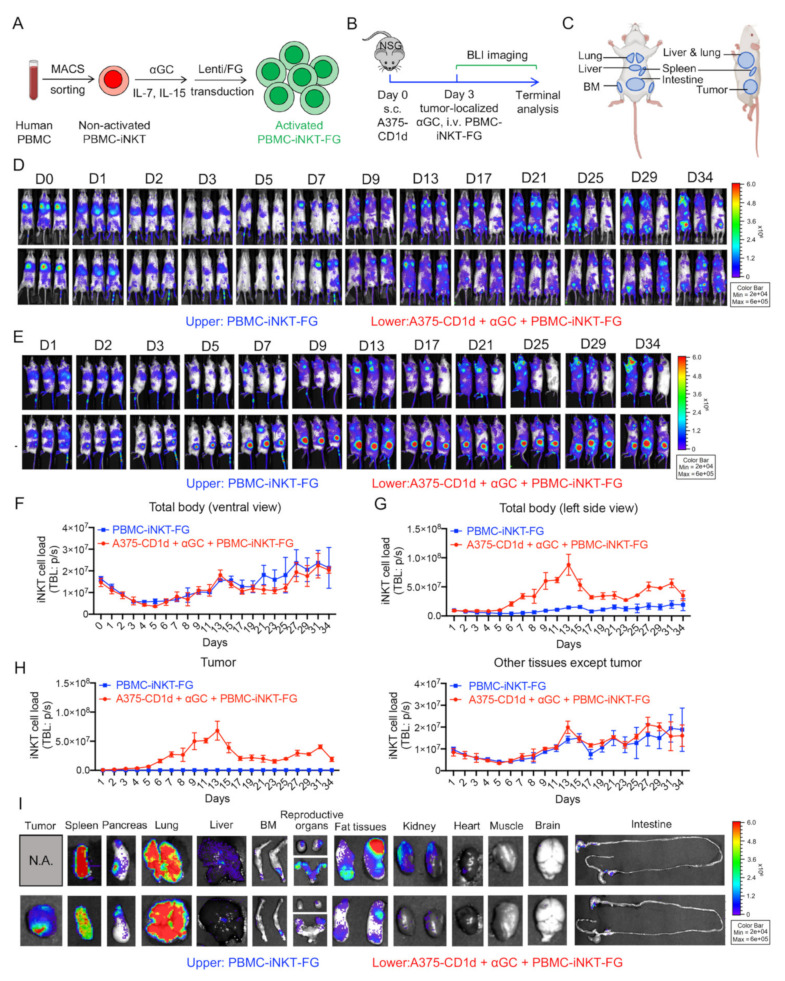
In vivo pharmacokinetics/pharmacodynamics (PK/PD) study of PBMC-iNKT cells in an A375-CD1d human melanoma xenograft NSG mouse model. (**A**) Generation of PBMC-iNKT cells overexpressing FG. (**B**) Experimental design. (**C**) The templates showing the distribution of tissues in the experimental mice from the ventral and left side views. BM, bone marrow. (**D**) BLI images showing iNKT cell load in experimental mice over time. The mice were imaged from the ventral view. The upper panel shows the experimental mice injected with PBMC-iNKT-FG cells, and the lower panel shows the experimental mice injected with A375-CD1d tumor cells, αGC, and PBMC-iNKT-FG cells. (**E**) BLI images showing iNKT cell load in experimental mice over time. The mice were imaged from the left side view. (**F**) Quantification of (**D**) (*n* = 3). (**G**) Quantification of (**E**) (*n* = 3). (**H**) Quantification of iNKT cell load in tumor and tissues in experimental mice over time. The quantification data of other tissues were obtained by subtracting iNKT cell load in tumor from that in total body of experimental mice. (**I**) BLI images showing iNKT cell load in multiple tissues from experimental mice on day 35. Representative of 2 experiments.

## Data Availability

Data supporting reported results are available on request from the corresponding author.

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
