# Peer review of "Tumor-Localized Administration of α-GalCer to Recruit Invariant Natural Killer T Cells and Enhance Their Antitumor Activity against Solid Tumors"

_ijms, 2022, doi:10.3390/ijms23147547_

Round 1

Reviewer 1 Report

The original article "Tumor-localized administration of α-GalCer to recruit invariant natural killer T cells and enhance their antitumor activity against solid tumors", presented by Yan-Ruide Li et al., hypothesizes about the use of the synthetic glycolipid α-GalCer to activating, recruit, and infiltrate iNKT into solid tumors. For this, the researchers use a humanized NSG mouse model and three different approaches to determine tumor targeting, biodistribution, and therapeutical benefits of the activation of iNKT cells by α-GalCer on lung and melanoma cancer.

This work shows a clear determination of what we want to analyze and focuses immediately on the main assays, indicating a previous in vitro work done by the lab members.

I think the approaches are adequate but still miss some more data to improve or complete the assays done.

Minor points:

1- BLI images are small, mainly in figure 5. One cannot distinguish tissue distribution in such small images. I know it is a hug panel showing very much and important information, but these BLI images deserve more.

2- In the discussion section you mention that “iNKT cell-based therapies has become an area of significant interest in recent years; multiple relevant” (lines 243-4), and you refer to a review from 2017. Searching by “iNKT cell”, only at the NIH (not other official agencies), appear 29 different human trials in which 7 are complete and 3 of them have results. I think that a brief summary of the main results obtained would enhance the importance of these new therapeutical approach.

3. The activation of iNKT cells, a very small subpopulation of Tcells, it is supposed to change some expression levels and it would also alter the surrounding cells like endothelial cells. Have you tested whether conditional medium form iNKT cells would change/activate endothelial cells (ECs)? A way to demonstrate the lack of effects on ECs is to develop capillary-like formation assays (Matrigel tubulogenesis assays) and viability or proliferation assays.

4- About the in vivo assays. I think the approaches are excellent, but I think you could squeeze a bit more the experiments.

4.1- You talk about tumor surface and the formula used is Width x Length mm2. When is a subcutaneous xenograft, surface volume can hide the real tumor growth. It is quite easy to calculate the tumor volume using a very similar formula: Width2 x Length x 0.52 (mm3). Therefore, I suggest showing tumor volume and not tumor surface.

4.2- Tumor growth kinetics are very good but on the last days of the assay, tumors start to grow (in the lung carcinoma assay). Do you think that a second dose of α-GalCer would enhance the antitumor response of iNKT cells?

4.3- Even that the primary tumor response is almost complete, you know that metastases are the main cause of death in most of the solid tumors. Have you checked for metastases? Since the tumors are GFP-Luc positive cells, it would be important to determine (GFP expression or IHC), the lack of metastases (classically in lungs and mor difficult in bone, brain or liver).

4.4- Since you are boosting/triggering the Immune System and it is correlated with inflammation processes that can be used in favor of the tumor progression and/or metastases. Have you tested for changes on immune parameters such as leukocyte cell numbers, cytokines serum levels; or angiogenic proteins like as VEGF, MMPs?

Reviewer 2 Report

In this manuscript Yan-Ruide Li, et al. described results of their work aimed at evaluation of effects of tumor-localized administration of alpha-galactosylceramide (α-GalCer), a specific ligand activating invariant natural killer T (iNKT) cells, on antitumor activity of these cells against solid tumors. This work is a continuation of a large series of studies by this team devoted to preclinical trials of the possibilities of using iNKT cells for antitumor immunotherapy. This field of investigation is significant because in recent years many publications have appeared that testify to the prospects of this approach to anticancer therapy.

For the study the authors used populations of enriched and activated human iNKT cells interacting with two lines of human cancer cells in in vitro and mouse in vivo models. The analysis of these interactions was conducted with application of modern adequate methods. The investigation revealed that combination of iNKT cell-based adoptive therapy with tumor-localized α-GalCer inoculation had high significant effect for solid cancer treatment.

In general the study is well presented, proper controls are used and the conclusions are convincingly supported by experimental results, the data are of considerable novelty and interest. Manuscript is well written.

Two minor suggestions might improve the overall quality of the manuscript:

1. Page 2, line 86, 87. “ … then stimulated with α-Galactosylceramide (αGC)-loaded do-86 nor-matched PBMCs (Figure 1A). …” It should be more correct (more understandable) to add “irradiated PBMCs”.

2. Page 11, line 405. “ … engineered PBMC-iNKT cells were stained using flow cytometry …”, probably not “stained” but “sorted” ...

Round 2

Reviewer 1 Report

Dear authors.

Some of the changes donde in the mansucritp are unable to be seen, e.g., Revised Figure 1. even taht they appear in the rebutal letter.

Therfore, I cannot accept the manuscritp like this. Please, include it and remove the comments that appear (but also unable to de be read) in the second version.

Author Response

Dear Reviewer, Thank you for the suggestion. We think the Dara from Revised Figure 1 (Figure R1) doesn't quite fit the current manuscript. Therefore, We put the data in the rebuttal letter but didn't incorporate it in the manuscript. We'd like not to include the figure R1 in the manuscript. We have consulted with journal editor for the decision